# Cellular neuromodulation in artificial networks

**Nicolas Vecoven** *
University of Liège

**Damien Ernst**
University of Liège

**Antoine Wehenkel**
University of Liège

**Guillaume Drion**
University of Liège

## Abstract

Animals excel at adapting their intentions, attention, and actions to the environment, making them remarkably efficient at interacting with a rich, unpredictable and ever-changing external world, a property that intelligent machines currently lack. Such adaptation property strongly relies on *cellular neuromodulation*, the biological mechanism that dynamically controls neuron intrinsic properties and response to external stimuli in a context dependent manner. In this paper, we take inspiration from cellular neuromodulation to construct a new deep neural network architecture that is specifically designed to learn adaptive behaviours. The network adaptation capabilities are tested on navigation benchmarks in a meta-learning context and compared with state-of-the-art approaches. Results show that neuromodulation is capable of adapting an agent to different tasks and that neuromodulation-based approaches provide a promising way of improving adaptation of artificial systems.

## 1 Introduction

We are now seeing the emergence of highly efficient algorithms that are capable of learning and solving complex problems. However, it remains difficult to learn models that generalise or adapt themselves efficiently to new, unforeseen problems based on past experiences. This calls for the development of novel architectures specifically designed to enhance adaptation capabilities of current deep neural networks (DNN).

In biological nervous systems, cellular neuromodulation provides the ability to continuously tune neurons input/output behavior to shape their response to external inputs in different contexts, generally in response to an external signal carried by biochemicals called neuromodulators [2, 9]. Neuromodulation regulates many critical nervous system properties that cannot be achieved solely through synaptic plasticity [7, 8], which represents the ability for neurons to tune their connectivity during learning. Neuromodulation has been shown to be critical to the adaptive control of continuous behaviours, such as in motor control among others [7, 8]. We propose a new neural architecture specifically designed for DNNs and inspired from cellular neuromodulation which we call NMN, standing for "Neuro-Modulated Network".

At its core, the NMN architecture is made of two neural networks: a main network and a neuromodulatory network. The main network is a feed-forward DNN composed of neurons equipped with a parametric activation function specifically designed for neuromodulation. It allows the main network to be adapted to new unforeseen problems. The neuromodulatory network, on the other hand, controls the neuronal dynamics of the main network via the parameters of its activation functions. Both networks have different inputs: whereas the main network is in charge of processing samples, the neuromodulatory network processes feedback and contextual data.

In [11], the authors take inspiration from Hebbian plasticity to build networks with plastic weights, allowing them to tune their weights dynamically. In [10] the same authors extand their work by learning a neuromodulatory signal that dictates which and when connections should be plastic. Our architecture is also related to hypernetworks [5], in which a network's weights are computed through another network. Other recent works focused on learning fixed activation functions [1, 6].

---

*nvecoven@uliege.be

33rd Conference on Neural Information Processing Systems (NeurIPS 2019), Vancouver, Canada.

## 2 NMN

The NMN architecture revolves around the neuromodulatory interaction between the neuromodulatory and main networks. We mimick biological cellular neuromodulation [3] in a DNN by assigning the neuromodulatory network the task to tune the slope and bias of the main network activation functions.

Let $\sigma(x) : \mathbb{R} \to \mathbb{R}$ denote any activation function and its neuromodulatory capable version $\sigma_{\text{NMN}}(x, \mathbf{z}; \mathbf{w}_s, \mathbf{w}_b) = \sigma\left(\mathbf{z}^T(x\mathbf{w}_s + \mathbf{w}_b)\right)$ where $\mathbf{z} \in \mathbb{R}^k$ is a neuromodulatory signal and $\mathbf{w}_s, \mathbf{w}_b \in \mathbb{R}^k$ are two parameter vectors of the activation function, respectively governing a scale factor and an offset. In this work, we propose to replace all the main network's neurons activation function with their neuromodulatory capable counterparts. The neuromodulatory signal $\mathbf{z}$, which size $k$ is a free parameter, is shared for all these neurons and computed by the neuromodulatory network as $\mathbf{z} = f(\mathbf{c})$. The function $f$ can be any DNN taking as input the vector $\mathbf{c}$ representing some contextual inputs (e.g. $\mathbf{c}$ may have a dynamic size in which case $f$ would be parameterized as a recurrent neural network (RNN) or a conditional neural process [4]). The complete NMN architecture and the change made to the activation functions are depicted on Figure 1.

Notably, the number of newly introduced parameters scales linearly with the number of neurons in the main network whereas it would scale linearly with the number of connections between neurons if the neuromodulatory network was affecting connection weights, as seen for instance in the context of hypernetworks [5]. Therefore this approach can be extanded to very large networks.

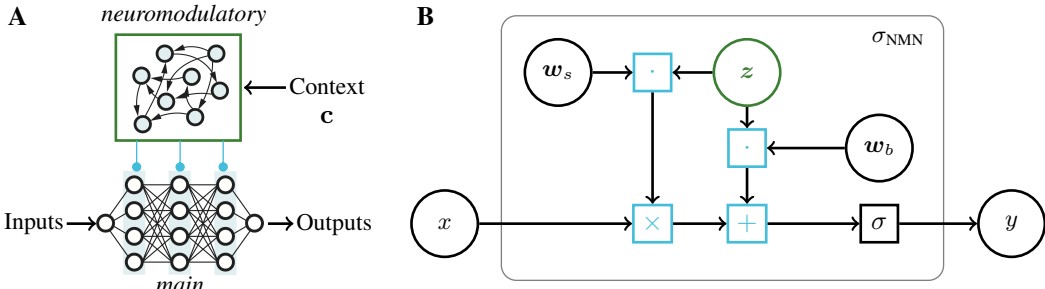

Figure 1: NMN architecture. **A.** The interaction of a *neuromodulatory* neural network (top) and a *main* neural network (bottom). **B.** Computation graph of the NMN activation functions.

## 3 Experiments

**Setting.** We evaluate the NMN architecture on meta-RL [14] which is motivated by the analogy with biology [10]. In contrast with classical RL, which is formalized as the interaction between an agent and an environment defined as a markov decision process (MDP), the meta-RL setting resides in the sub-division of an MDP as a distribution $\mathcal{D}$ over simpler MDPs. Let $t$ denote the discrete time, $\mathbf{x}_t$ the state of the MDP at time $t$, $\mathbf{a}_t$ the action taken at time $t$ and $r_t$ the reward obtained at the subsequent time-step. At the beginning of a new episode $i$, a new element is drawn from $\mathcal{D}$ to define an MDP, referred to by $\mathcal{M}$, with which the meta-RL agent interacts for $T \in \mathbb{N}$ time-steps afterwards. The only information that the agent collects on $\mathcal{M}$ is through observing the states crossed and the rewards obtained at each time-step. We denote by $\mathbf{h}_t = [\mathbf{x}_0, \mathbf{a}_0, r_0, \mathbf{x}_1, \ldots, \mathbf{a}_{t-1}, r_{t-1}]$ the history of the interaction with $\mathcal{M}$ up to time step $t$. As in [14], the goal of the meta-learning agent is to maximise the expected value of the sum of rewards it can obtain over all episodes and steps.

In [14], the authors tackle this meta-RL framework by using an advantage actor-critic (A2C) algorithm, in which the actor and the critic are RNNs, taking $[\mathbf{h}_t, \mathbf{x}_t]$ as input. In this work, we propose to compare the NMN architecture to standard RNN by modelling both the actor and the critic with NMN. To this end, we define the feedback and contextual inputs $\mathbf{c}$ (i.e. the neuromodulatory network inputs) as $\mathbf{h}_t$ while the main network's input is defined as $\mathbf{x}_t$. Note that $\mathbf{h}_t$ grows as the agent interacts with $\mathcal{M}$, motivating the usage of a RNN as neuromodulatory network. To be as close as possible to the neuronal model proposed by [3], the main network is a fully-connected neural network built using saturated rectified linear units activation functions $\sigma(x) = \min(1, \max(-1, x))$, except for the final layer (also neuromodulated), for which $\sigma(x) = x$.

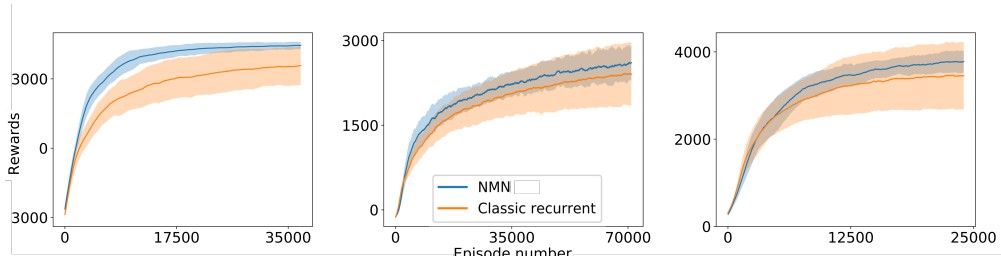

Figure 2: Mean (± std in shaded) sum of rewards obtained over 15 training runs with different random seeds with respect to the episode number. Results of benchmark 1,2 and 3 are displayed from left to right. The plots are smoothed thanks to a running mean over 1000 episodes.

We build our models such that both standard RNN and NMN architectures have the same number of recurrent layers/units and a relative difference between the numbers of parameters that is smaller than 2%. Both models are trained using an A2C algorithm with generalized advantage estimation [12] and proximal policy updates [13]. Finally, no parameter is shared between the actor and the critic.

**Benchmarks.** We carried out our experiments on three custom benchmarks: a simple toy problem and two navigation problems with sparse rewards. These benchmarks are built to evaluate our architecture in environments with continuous action spaces. For conciseness and clarity, we only provide a mathematical definition of the first benchmark (later needed for discussing results). The two other benchmarks are briefly textually depicted and further details are available on Github [2].

We define the first benchmark (made of a 1D state space and action space) through a random variable $\alpha$, informative enough to distinguish all different MDPs in $\mathcal{D}$. With this definition, $\alpha$ represents the current task and drawing $\alpha$ at the beginning of each episode amounts to sample a new task in $\mathcal{D}$. At each time-step, the agent observes a biased version $x_t = p_t + \alpha$ of the exact position of a target $p_t \in [-15, 15]$, with $\alpha \sim \mathcal{U}[-10, 10]$. The agent takes an action $a_t \in [-20, 20]$ and receives a reward $r_t$ which is equal to 10 if $|a_t - p_t| < 1$ and $-|a_t - p_t|$ otherwise. In case of positive reward, $p_{t+1}$ is re-sampled uniformly in its domain else $p_{t+1} = p_t$.

The second benchmark consists in navigating towards a target in a 2D space with noised movements. At each time-step, the agent observes its relative position to the target and outputs the direction of a move vector $\mathbf{m}_t$. A perturbation vector $\mathbf{w}_t$ is then sampled uniformly in a cone, whose main direction is dictated by the current task in $\mathcal{D}$. Finally the agent is moved following $\mathbf{m}_t + \mathbf{w}_t$. If the agent reaches the target, it receives a high reward and is moved to a position sampled uniformly in the 2D space.

The third benchmark also consists in navigating in a 2D space but containing two targets. At each time-step the agent observes its relative position to the two targets and is moved along a direction given by its action. In this benchmark, $\mathcal{D}$ is only composed of two tasks, corresponding to the attribution of a positive reward to one of the two targets and a negative reward to the other. As for benchmark 2, once the agent reaches a target, it receives the corresponding reward and is moved to a position sampled uniformly in the 2D space.

**Results.** From a learning perspective, a comparison of the sum of rewards obtained per episode by NMNs and RNNs on the three benchmarks is shown on Figure 2. The results show that in average, NMNs learn faster (with respect to the number of episodes) and converge towards better policies than RNNs (i.e., higher rewards for the last episodes). Most notable, NMNs show very stable results, with small variances over different random seeds, as opposed to RNNs.

From an adaptation perspective, Figure 3 shows the temporal evolution of the neuromodulatory signal $\mathbf{z}$ (part **A**) and of the rewards (part **B**) obtained with respect to $\alpha$ for 1000 episodes played on benchmark 1. For small values of $t$ the agent has little information on the current task, leading to a non-optimal behavior (as it can be seen from the low rewards). Most interestingly, the signal $\mathbf{z}$ for the first time-steps exhibits little dependence on $\alpha$, highlighting the agent's uncertainty on the current task. Said otherwise, for small $t$ the agent learnt to play a (nearly) task-independent strategy. As time passes, the agent gathers further information about the current task and approaches a near-optimal policy. This reflects in convergence of $\mathbf{z}$ with a clear dependency on $\alpha$ and also in wider-spread values of $\mathbf{z}$. For large value of $t$, $\mathbf{z}$ holding constant

---

[2]`https://github.com/nvecoven/nmd_net`

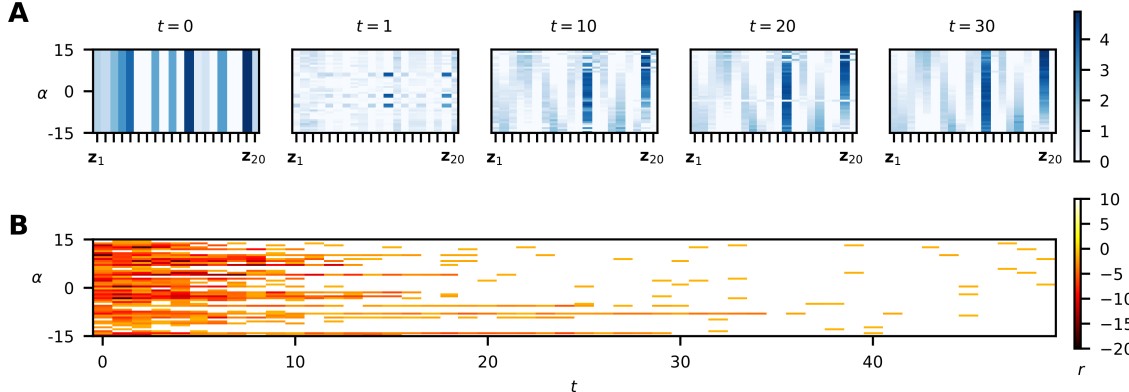

Figure 3: Adaptation capabilities of the NMN architecture on benchmark 1. **A.** Temporal evolution of the neuromodulatory signal $\mathbf{z}$ with respect to $\alpha$, gathered on $1000$ different episodes. **B.** Rewards obtained at each time-step by the agent during those episodes.

between time-steps shows that the neuromodulatory signal is almost state-independent and serves only for adaptation. Finally, we note that the value of $\mathbf{z}$ in each of its dimensions varies continuously with $\alpha$, meaning that for two similar tasks, the signal will converge towards similar values.

## 4 Conclusions

In this work, we use a high level view of a nervous system mechanism called cellular neuromodulation to improve artificial neural networks adaptive capabilities. The results obtained on three meta-RL benchmark problems showed that this new architecture was able to perform better than classical RNN. The work reported in this paper could be extended along several lines. First, it would be interesting to explore other types of machine-learning problems where adaptation is required. Second, research work could also be carried out to further improve the NMN introduced here. For instance, one could introduce new types of parametric activation functions which are not linear, or spiking neurons. It would also be of interest to look at sharing activation function parameters per layer. Furthermore, analysing more in-depth the neuromodulatory signal (and its impact on activation functions) with respect to different more complex tasks could also be worth-while. Finally, let us emphasize that even if the results obtained by our NMN are good and also rather robust with respect to a large choice of parameters, further research is certainly still needed to better characterise their performances.

### Acknowledgments

The authors would like to acknowledge Gilles Louppe and Antonio Sutera for useful feedback on the manuscript. Nicolas Vecoven has a grant funded by the FRIA Belgium and acknowledges its financial support. Antoine Wehenkel is a research fellow of the F.R.S.-FNRS (Belgium) and acknowledges its financial support.

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
