# OpenReview forum: "Cellular neuromodulation in artificial networks"
_NeurIPS.cc/2019/Workshop/Neuro_AI — Real Neurons & Hidden Units @ NeurIPS 2019 Poster_

### Official Review · AnonReviewer2 · 2019-09-24
**Clear enough & relevant, if incremental**

**Clarity:** 3

**Comment:**

I'd say a fairly 'standard' work for the setting. Only real point for improvement is more earnest bench marking/model comparison. Authors could also add some context by considering related works in the computational neuroscience literature, e.g. Stroud et al. Nature Neurosciencevolume 21, pages 1774–1783 (2018) and https://arxiv.org/abs/1902.05522 (though the latter is very recent).

**Category:**

Neuro->AI

**Clarity Comment:**

Paper is clear and quite readable.

**Evaluation:**

3: Good

**Importance:**

2: Marginally important

**Importance Comment:**

The proposed model is essentially a constrained/specific parameterisation within the broader class of 'context dependent' models. The heavy lifting is seemingly done by well known architectures: default RNN & a feed-forward NN.
While it does not seemingly add anything conceptual, the exact implementation is arguably new.

**Intersection:**

4: High

**Intersection Comment:**

The paper takes a crudely 'neuroscience inspired' concept (though, admittedly it could simply be 'task structure' inspired) and builds a simple model from it, which it benchmarks on a appropriately designed simplest-working-example. So it fits well with the workshop theme.

**Rigor Comment:**

The model description is nice and clear. I think a more persuasive bench marking could be done. Perhaps compare to reference models [11] or [10] rather than a 'vanilla' RNN, as this amounts to not using any prior information about the task (which, by construction, we 'know' is useful).
Also perhaps report results from one of the 2 (mentioned) more complex benchmarks.

**Technical Rigor:**

2: Marginally convincing

---

### Official Review · AnonReviewer3 · 2019-09-25
**Learned, context-dependent activation functions in Meta-RL tasks**

**Clarity:** 4

**Comment:**

Strengths:

The paper is clearly written, well justified, and model is rigorously tested.

Areas for improvement:
The results are modest and I would be keen to see how the approach scales to more difficult benchmarks. The method they choose clearly reduces the variance in rewards gained, which is interesting in of itself. I would like to see whether this holds up. Additionally, I would like to see how this method performs when context must be inferred by the agent.

**Category:**

Neuro->AI

**Clarity Comment:**

The paper is well-written, with clear figures and descriptions of the model, task, and results.

**Evaluation:**

3: Good

**Importance:**

3: Important

**Importance Comment:**

Its an open question in neuroscience what the purpose of neuromodulation is in learning and behaviour, given that neuroscientists know a lot about their effects on intrinsic properties of neurons. It's also difficult to develop RL agents that generalize across tasks well. This paper addresses these questions along a similar vein to recent approaches (Miconi et al., 2018, 2019).

**Intersection:**

4: High

**Intersection Comment:**

The paper introduces a neuroscience-inspired solution to training RL agents to a behaviourally relevant problem, therefore is well-positioned at the intersection of neuroscience and AI.

**Rigor Comment:**

The authors implement a reasonable interpretation of the effects of contextual neuromodulation on the intrinsic properties of neurons via a recurrent neural network influencing the gain of learned scale and bias of node activation functions. The benchmark chosen is simple, and the treatment of the problem is rigorously addressed running over many seeds.

**Technical Rigor:**

3: Convincing

---

### Official Review · AnonReviewer1 · 2019-09-26
**Novel structure for context-dependent Meta-RL**

**Clarity:** 4

**Comment:**

An interesting and novel structure for learning across multiple tasks.  The results show improvement on state-of-the-art challenging benchmarks.

Detailed strengths and weakness are above.

**Category:**

AI->Neuro

**Clarity Comment:**

The paper was mostly well explained.  I think somewhere early a general statement of the problem would have helped.  For example, in Section 2, I think it could have been made more clear what is "context" and what is the training data and what is desired goal.  How do we measure performance generalization.  Also in the training section, some of the details were difficult to follow.  But, that could be a result of the space.


**Evaluation:**

4: Very good

**Importance:**

3: Important

**Importance Comment:**

The paper presents a novel structure for neural networks that can generalize to new tasks.  The structure appears new where a first DNN computes some terms, z,  based on a context.  The term z is then applied to the weights in the layers in a second network.  This structure potentially allows learning across new tasks.  The methods are tested on a standard Meta-ML benchmark and appear to outperform state-of-the-art methods.

**Intersection:**

4: High

**Intersection Comment:**

The problem of how algorithms can learn to generalize well across multiple tasks
and use context is clearly central to both ML and neuroscience.  The paper makes a case that the algorithms are "inspired" by biological systems.  But, if the goal of the paper is to understand how true biological systems work, I think there needs to be more detail on how this architecture would map biologically.  But, that obviously is a very hard problem and the results here should still be extremely useful.

**Rigor Comment:**

The paper takes on very challenging state-of-the-art problems with a sophisticated network.  The tests against the bench marks is rigorously and thoroughly performed.


**Technical Rigor:**

4: Very convincing

---

### Decision · Program_Chairs · 2019-10-02

Accept (Poster)